# A Multi-Component Framework for the Analysis and Design of Explainable Artificial Intelligence

**Mi-Young Kim [1,2,*,†], Shahin Atakishiyev [1,†], Housam Khalifa Bashier Babiker [1,†], Nawshad Farruque [1,†], Randy Goebel [1,†], Osmar R. Zaïane [1,†], Mohammad-Hossein Motallebi [1,†], Juliano Rabelo [1,†], Talat Syed [1,†], Hengshuai Yao [3] and Peter Chun [4,5]**

1    XAI Lab, Department of Computing Science, Alberta Machine Intelligence Institute, University of Alberta, Edmonton, AB T6G 2E8, Canada; atakishi@ualberta.ca (S.A.); khalifab@ualberta.ca (H.K.B.B.); nawshad@ualberta.ca (N.F.); rgoebel@ualberta.ca (R.G.); zaiane@ualberta.ca (O.R.Z.); motalleb@ualberta.ca (M.-H.M.); rabelo@ualberta.ca (J.R.); talat@amii.ca (T.S.)
2    Department of Science, Augustana Faculty, University of Alberta, Camrose, AB T4V 2R3, Canada
3    Huawei Technologies, Edmonton, AB T6G 2E8, Canada; hengshuai.yao@huawei.com
4    Department of Electrical and Computer Engineering, University of Alberta, Edmonton, AB T6G 2E8, Canada; peter.chun@huawei.com
5    Huawei Technologies Canada, Markham, ON L3R 5A4, Canada
*    Correspondence: miyoung2@ualberta.ca
†    Current address: Department of Computing Science, University of Alberta, 2-21 Athabasca Hall, Edmonton, AB T6G 2E8, Canada.

**Abstract:** The rapid growth of research in explainable artificial intelligence (XAI) follows on two substantial developments. First, the enormous application success of modern machine learning methods, especially deep and reinforcement learning, have created high expectations for industrial, commercial, and social value. Second, the emerging and growing concern for creating ethical and trusted AI systems, including compliance with regulatory principles to ensure transparency and trust. These two threads have created a kind of "perfect storm" of research activity, all motivated to create and deliver any set of tools and techniques to address the XAI demand. As some surveys of current XAI suggest, there is yet to appear a principled framework that respects the literature of explainability in the history of science and which provides a basis for the development of a framework for transparent XAI. We identify four foundational components, including the requirements for (1) explicit explanation knowledge representation, (2) delivery of alternative explanations, (3) adjusting explanations based on knowledge of the explainee, and (4) exploiting the advantage of interactive explanation. With those four components in mind, we intend to provide a strategic inventory of XAI requirements, demonstrate their connection to a basic history of XAI ideas, and then synthesize those ideas into a simple framework that can guide the design of AI systems that require XAI.

**Keywords:** interpretation; explanation; explainable artificial intelligence; causal explanation; explainee-specific explanation; evaluation of explainable AI

## 1. Introduction

Fueled by a growing need for trusted and ethical artificial intelligence (AI) by design, the emergence of the last decade of machine learning is crowded with a broad spectrum of research on explainable AI (XAI). The general idea is to somehow develop methods that capture abstractions of the predictive models created by modern machine learning methods that not only respect the properties of those automatically constructed models but, in addition, provide a basis for explaining their predictions.

A significant portion of the challenge arises because of the rapid and eclectic flurry of activities in XAI which have exposed confusion and controversy about foundational concepts such as interpretability, explanation, and causality.

Perhaps confusion and disagreement is not surprising given the complexity of modern learning methods. For example, when some deep learning methods can build predictive models based on more than 50 million distinct parameters, it is not a surprise that humans will debate what has been captured (e.g., see [1]; (especially see Section 2, Chapter 9, The Limitations of Deep Learning.)). Note also the confusion regarding misconceptions on a specious trade-off between predictive accuracy and explainability (cf. [2]), which have precipitated scientific workshops to address such misconceptions (note the 32nd Conference on Neural Information Processing Systems (NeurIPS 2018), Workshop on Critiquing and Correcting Trends in Machine Learning, https://ml-critique-correct.github.io (accessed on 7 December 2018)). Another example of yet-to-be-resolved issues includes the strange anomaly where syntactic reduction of the parameter space of some deep learning-created models actually results in improved predictive accuracy (e.g., [3,4]). The reality is that consensus on the foundations for scientific understanding of general machine learning, and thus XAI, is not yet sufficiently developed.

Even though the long history of science and more recent history of scientific explanation and causality have considerable contributions to make (e.g., [5,6]), it seems as though the demand created by potential industrial value has induced a kind of brittleness in identifying and confirming a robust trajectory from the history of formal systems to their role in modern applications of AI. However, we believe that one can re-establish some important scientific momentum by exploiting what Newell and Simon's Turing Award paper ([7]) identified as the physical symbol systems hypothesis: "A physical symbol system has the necessary and sufficient means for general intelligent action" (p. 116).

This need not require that a reader have more than an undergraduate course in formal logical methods, including statistics and probability. The more challenging aspect is to motivate a commitment to resolving ambiguity regarding concepts such as interpretation, logical inference, abductive explanation, and causality.

So the challenge is to clarify connections between the recent vocabulary of XAI and their historical roots in order to distinguish between scientifically valuable history and potentially new extensions. In what follows, we hope to articulate and connect a broader historical fabric of concepts essential to XAI, including interpretation, explanation, causality, evaluation, system debugging, expressivity, semantics, inference, abstraction, and prediction.

To do so, we need to articulate a general, albeit incomplete, XAI framework, which is immediately challenged to be comprehensive enough to acknowledge a role for a broad spectrum of components, yet still be of sufficient conceptual value to avoid becoming a naïve long and unstructured survey of overlapping—sometimes competing—concepts and methods. So, our approach is to start with a simple structural relationship amongst four popular XAI properties, illustrated in Table 1. We hope that those four components identified help seed the reader's questions about how all other identified elaborations fit into the framework we articulate. For example, in the case of requiring components that represent explicit domain knowledge in an application, a representation language's formal expressibility will determine what can be written, but not obviate the need for such representations.

**Table 1.** Tabular representation of levels of explanation and the corresponding attributes.

| Attributes | Level 0 | Level 1 | Level 2 | Level 3 | Level 4 |
|---|---|---|---|---|---|
| Explicit explanation representation | | ✓ | ✓ | ✓ | ✓ |
| Alternative Explanations | | | ✓ | ✓ | ✓ |
| Knowledge of the explainee | | | | ✓ | ✓ |
| Interactive | | | | | ✓ |

There are several explicit concepts or components that are common amongst all XAI studies: (1) the requirement for an explicit knowledge representation to support

explanation, (2) the need for alternative explanations, (3) the capture and use of knowledge specific to the entity receiving the explanation, and (4) whether or not an explanatory system is interactive. These initial components provide the basis for an initial high-level analysis of XAI.

In the case of the need of explicit representations, the easiest understanding arises from noting that the construction of predictive models from deep learning are typically considered "black box" models. This is because there is no explicit knowledge representation that supports the production of explanations for human users who will desire some level of transparency to confirm trust. This is universally true, especially in the application areas of medicine and law, but as we will see below, it is not just humans that will require explanations for building trust.

Whether explicit knowledge of the predictive models is conducted after the construction of the predictive model (so-called "post-hoc" explanation) or concurrently during the construction of the predictive model (cf. [8]) is not the issue. Rather, the issue is access to semantically aligned representations that provide knowledge appropriate to explanation construction (see more below).

In the case of the need for alternative explanations, good summaries of the social and technical basis for explanation (e.g., [9]) note that alternative explanations are more likely to be able to convey the semantics of an explanation, like a good teacher's presentation of alternative explanation for a student or the existence of alternative proofs for theorems in a more technical setting. The hypothesis is that multiple alternative explanations of the same prediction improve trust of predictive systems.

In the case of (3), it is not just the idea of alternative explanations but that knowledge of the explanation receiver, or "explainee", should shape an explanation for a particular explainee, again like a good teacher provides better explanations with knowledge of his/her students.

Finally, like the whole history of Socratic dialogue and knowledge transfer, interaction between explainer and explainee helps identify and fill knowledge gaps that always exist in the conveyance of an explanation.

Note that most surveys of XAI note the requirement for a system to produce alternative explanations; simply put, producing a single explanation may be completely insufficient for multiple explainees [9]. Similarly, many have noted the value of interactive XAI systems and dialogue systems [10], which provide a basis for an explainee to submit and receive responses to questions about a model prediction, including alternative explanations, and thus build deeper trust of the system.

Further note that these four central concepts informally suggest the need to somehow identify the quality of explanations. That idea is more fully explored below and, in fact, throughout our paper.

To start, we need to distinguish what we consider a kind of quality of explanatory system, with the intuition that explanations can be confirmed as "better" by some evaluation measure, e.g., we expect a causal explanation to provide the basis for recreating a causal inference chain to a prediction. So, for example, in the case of a medical diagnostic prediction based on observed symptoms, a causal explanation would align with biological knowledge about causation. Alternatively, in much weaker and certainly non-causal explanations, we would distinguish "explanation by authority" towards the simpler end of a quality scale; for example, a question such as "Why can't I go out tonight?" may have an explanation from a parent as "Because I said so," which we might just say is explanation by authority. Just these simple distinctions help frame and motivate a more detailed analysis of distinctions across our informal explanatory framework, which will be further articulated below. We provide increasing detail and precision about connections to existing scientific ideas and about how we believe existing XAI concepts align with this simple framework. This, we hope, will inform how to articulate choices in the design of an XAI system for use in any AI application. In fact, we believe that a significant measure of an XAI system's production of explanations will be based on subject matter expertise in

application (e.g., [11–14]). Our contributions can be summarized as follows: (1) we provide a strategic inventory of XAI requirements, (2) we demonstrate a connection between XAI requirements to a history of XAI ideas, and (3) we synthesize the ideas of XAI requirements into a simple framework to inform XAI system design and XAI system evaluation.

The rest of our paper is organized as follows. Section 2 presents further details on what we consider the principal components of XAI, including that explanations need to be explainee specific, that there can always be multiple explanations, and that the evaluation of the quality of explanation has more to do with the explainee than the explanation system itself. This will help provide sufficient detail to articulate the relationship between current explanatory concepts and their relationship to historical roots, e.g., to consider the emerging demands on the properties of a formal definition of interpretability by assessing the classical formal systems view of interpretability. Section 3 considers the more general history of explanation as an attempt to connect the formal philosophy and scientific explanation foundations. This concludes with the articulation of the explanatory role of recent theories of causal representation. Section 4 summarizes important emerging trends and components in proposed XAI architectures, including those that apply to both machine-learned predictive models and general AI systems. Section 5 provides a synopsis of current XAI research threads and how they might integrate into an emerging XAI architecture. We include the description of important XAI ideas such as pre-hoc versus post-hoc explanation creation, and the evaluation of explanations, in order to sketch an architecture of how necessary components for XAI are connected. Finally, Section 6 provides a brief summary, and what we believe the future architectures of XAI systems will require to ensure the trust of future AI systems.

## 2. Principal Components at the Foundations of XAI

### 2.1. Explainability and Interpretability

The current literature harbors some confusion regarding the definitions of interpretability and explainability of models. While many recent papers use those terms interchangeably [15,16], some other papers do make a distinction between those terms, but these too are unclear. For example, Gilpin et al. [17] define interpretability as a rudimentary form of explainability. Rudin [18] finds that there is no single definition on interpretability. However, she defines a spectrum which extends from fully interpretable models, such as rule-based models (that provide explanations by definition), to deep models that cannot provide explanations out of the box.

We note that there is no confusion about interpretation, explainability, and semantics in the case of the history of mathematical logic (e.g., [19]). When the vocabulary of the representation (well-formed formulae) is precise, interpretability is obtained by ensuring that each component is assigned a fixed interpretation (e.g., constants to individuals in a world, variables range over constants, truth values to logical connectives, etc.). Additionally, the semantic interpretation of any expression is determined compositionally by interpretation of an expression's components.

However, the manner in which representations emerge in the context of empirical developments in machine learning has not typically been guided by any adaptation of extension of the systems of interpretability and semantics of logic. Our perspective is that the principles of mathematical logic can be easily adopted to a broad range of machine learned representations in order to support XAI from explicit learned representations. Note further that formal logical concepts include a formal basis for probability and statistics, so our analysis does not exclude the newly labeled "neurosymbolic" systems [20]

In this context, an interpretable model is one that an explainee can read or inspect and analyze in terms of composable parts. In this way, interpretability refers to a static property of the model and can vary from fully interpretable (for models such as small decision trees) to deep neural network models in which interpretability is more complex and typically limited. For instance, consider what each layer learns in a convolutional neural network (CNN) for image analysis: early layers are responsible for extracting low-

level features such as edges and simple shapes, while later layers usually extract high-level features whose semantics are understood with respect to an application domain. In fact, with this perspective, models such as deep neural networks could hardly be classified as interpretable. It is important to point out that interpretability applies to a learned model before considering the inference the model can conduct. Note that we are against classifying models as interpretable or non-interpretable, but rather we believe there should be a spectrum allowing an interpretability measure to be assigned to each model.

On the other hand, explainability has to deal with what kind of output the system provides to an explainee rather than how that explainee directly interprets the meaning of each model component. In other words, explanation has to do with clarifying the reason or reasons a prediction was made or an action was taken. So, we define an explainable model as a system which is capable of providing explanations by detailing/recording its own prediction process (e.g., in a rule-based system, this would be the sequence of rules which were fired by the given input). Explainability is, thus, a dynamic property of a model since that model requires run-time information to produce explanations for its output. Explainability pertains to the mechanism of detailing/justifying an inference or prediction using a learned model, whether the used model is clearly interpretable or loosely interpretable. Figure 1 illustrates the distinction between interpretability, which concerns the rendition or comprehension of a predictive model learned from data during training, and explainability, which pertains to the elucidation and justification of a prediction or decision made in the presence of a new observation or case. Both may revert to and rely on the original training data for analogy or as a basis for justification.

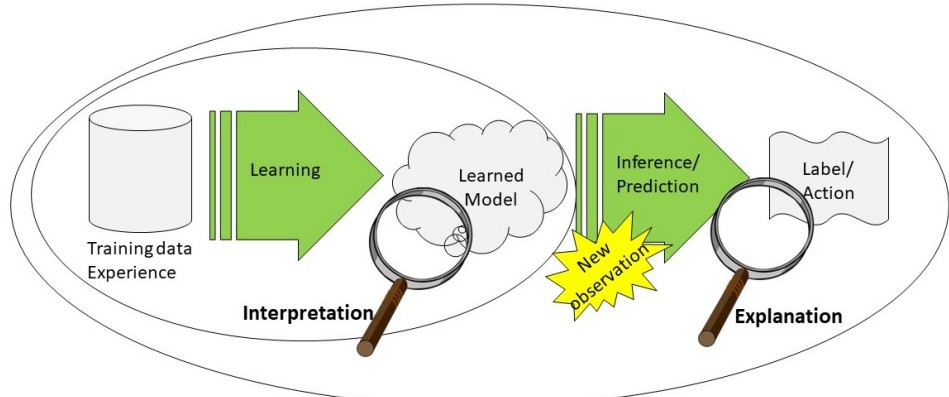

**Figure 1.** Interpretability of a model vs. explainability of a prediction.

Based on the above definitions, models such as small decision trees and rule-based systems are fully-interpretable and explainable, while deep models cannot be considered either. Once an explanation module is added to a deep neural model (e.g., [21]) it becomes an explainable system, but interpretability remains an issue: given the corresponding training data, one can investigate and understand how the model operates, hypothesize on what could be undertaken to improve its performance, and even devise techniques to improve the learning mechanism. However, interpretation, as per our definition, of a learned deep neural net is not possible.

Notice that these distinctions between explainability and interpretability do not comply with a reasonable assumption that is true in a common-sense usage of the terms. Outside of the AI arena, it is reasonable to expect that explainability requires interpretability; a person can only provide explanations for something they stated (or predicted) if they fully understand their mental models and can interpret them to establish which facts were considered and which reasoning process took place. As per our definition of explainability in AI, an explanation carries a completely different meaning from that of its usual usage and does not require interpretability, i.e., a model can be explainable but not interpretable. An example of that is a saliency map as an explanation of a prediction made by a CNN: the

highlighted areas of an image can be considered an explanation of an image classification system. That "explanation" does not consider image semantics, it is just an objective identification of which pixels contribute more to the final activations in a neural network. Still, they can be especially useful as debugging tools for machine learning practitioners (see Section 2.3).

### 2.2. Alternative Explanations: Who Are Explanations for?

According to [22], a person's background knowledge, often called prior knowledge, is a collection of "abstracted residue" that has been formed from all of life's experiences and is brought by everyone, at any age, to every subsequent life experience, as well as being used to connect new information to old.

As such, it becomes clear that, in the context of XAI, systems should be able to effectively take background knowledge into consideration in order to connect predictions and predictive models and to shape explanations to the appropriate level of detail, i.e., adjusting explanations to conform to the knowledge of the corresponding explainee. However, the most common current approaches to explainability in AI systems attempt to provide information on a model's inner functioning without regard for the consumer of that information (see Section 2.3).

To illustrate the importance of considering the explainee (and hence their background knowledge, expectations, etc.), consider an interview by Richard Feynman with the British Broadcasting Corporation (BBC) in 1983, in which he was asked why magnets with the same poles repel each other when placed close enough [23]. Feynman argued that, to properly explain that behaviour, he would need to consider the reporter's background on that matter and any answer he provided could unfold a new round of questions and explanations; this process could continue indefinitely, as new details are provided to explain the previous answer. The point is, of course, that an explainee's satisfaction with this iterative dialogue is the foundation of how XAI systems should be evaluated (cf. Sections 2.2 and 5.3).

### 2.3. Debugging Versus Explanation

As mentioned above, some approaches to explainability provide information related to how the model works internally. However, not all information provided by those approaches can really be considered explanatory information. Of course, the information provided by, e.g., rule-based systems, can be understood as a detailed explanation on how the system operates and could be useful in scenarios where an end user needs to understand how the system generated a prediction. However, other approaches (especially those applicable to the so called opaque or black-box systems) are much less informative and can be considered superficial "hints" rather than actual explanations, e.g., saliency maps on convolutional neural networks. This is really an observation about understanding internal components so as to debug those mechanisms rather than an explanation of a prediction. Although semantically weak from an application domain perspective, these approaches are still helpful to understand how a model behaves, especially if the consumer of the information has the necessary background. Hence, in general, this kind of explanation may be considered as debugging support for opaque AI systems rather than production of explanations based on a user's understanding of the semantics of an application domain.

One example of a debugging tool to augment a model is the work of [24], in which the authors used a ResNet [25] to recognize objects in a scene. Applying a saliency map to determine what area in the image contributes to the final activations is not really helpful for a (lay) human consuming the model output to understand misclassifications. However, this may help a programmer to figure out alternatives to overcome model limitations. In this case, the authors augmented the model by post-processing the final results using an external knowledge graph to provide semantic context and modify the confidence score of the recognized objects.

An alternative, perhaps more foundational model, is presented by Evans and Greffenstette in [26], who articulate an inductive logic programming (ILP) framework in which

explanatory models are identified in the space of all possible inductive logic programs. This framework requires the development of a measure space for all such ILP instances, to provide a gradient in that space. However, the positive consequences of that technical maneuver is that any instance of an inductive logic program can be interpreted at the level of the semantics of an application domain, if necessary, all the way down to instructions for a Turing machine. This framework does not resolve the challenge of what an appropriate level of explanation should be for a particular explainee, but it does provide a rich and mathematically elegant space in which to identify descriptions of computation to arrive at a predictive model all the way to rule-based specifications at the level of an application domain.

### 2.4. Is There a Trade-off between Explanatory Models and Classification Accuracy?

Deep learning-based systems have become prevalent in AI, especially after successful applications in image classification problems. Deep learning-based systems achieve impressive accuracy rates on standard datasets (e.g., ImageNet [27]) without requiring much effort on designing and implementing handcrafted rules or feature extractors. In fact, by leveraging transfer learning techniques and well-known architectures based on convolutional neural networks, a deep learning practitioner can quickly build an image classifier that outperforms image classification methods which were state of the art before the "deep learning revolution."

Nevertheless, despite their excellent overall performance, deep learning systems are considered black-boxes and unable to provide explanations as to why they make a given prediction. In some applications, that limitation does not translate into a serious practical problem: a mobile phone picture classification application which misclassifies two animals will not bring consequences to a user other than a few giggles and an amusing discussion topic. If those errors are seldom, nobody would really care or lose confidence in the application. A canonical example is a panda picture being classified as a gibbon after some adversarial noise is added (see Figure 1 in [28]). However, the referred example illustrates it is possible to intentionally fool a classifier through addition of appropriate noise. Depending on the image classification application, that kind of error may produce more serious consequences than the hypothetical phone application mentioned above. For example, recently McAfee Advanced Threat Research (ATR) hackers were able to fool Tesla's autopilot by tampering speed limit signs with adhesive tape, enabling the car to accelerate beyond the actual speed limit in that area [29]. This simple example illustrates that predictions from AI models cannot be blindly accepted in many practical applications. Moreover, techniques unable to explain how they arrive at a prediction make them even more sensitive to random errors or deliberate attacks. That observation raises an important question around a potential trade-off between model accuracy and explanatory capabilities. It is true that a deep learning-based model can achieve accuracy in many practical applications and allows practitioners to quickly build accurate models with little effort. However, some preconditions do exist, the main one being the availability of potentially large labeled datasets (a problem that can be alleviated by transfer learning, but still common in machine learning in general and in deep learning techniques in particular). In some cases, training large state-of-the-art deep learning networks requires thousands or even millions of dollars (the estimated cost of training just one of the models developed in [30] was estimated at USD 1.4 million [31]). All considered, it is not appropriate to claim there is necessarily a trade-off between accuracy and explainability (or, more generally, model performance). In some cases, deep learning methods will not be able to provide state-of-the-art results (e.g., when there is not enough labeled data, when the model is so large it will be impractical to deploy on the target platforms or even train due to prohibitive costs, etc.) so more explanation-capable techniques might even provide better results. However, as previously noted, there is no reason in principle that induced models, such as decision trees, should be less accurate than deep-learned models.

## 2.5. Assessing the Quality of Explanations

Whereas a factually wrong explanation is obviously inappropriate, determining if an explanation is good transcends its correctness. The quality of an explanation is a little like beauty—it is in the eye of the beholder. It is very clear (and quite common) that two factually correct, but different explanations could be considered good or bad depending on to whom they were provided. This means that, to assess quality of explanations, one (again) needs to consider the explainee, the person who receives the explanation (see Section 2.2). The explainee's background, expectations, goals, context, etc., will play a determinant role in the evaluation process.

From the above, it is clear that assessing the quality of explanations has a subjective component and is a quite complicated task, even if performed manually. So, defining an effective technique to evaluate explanation capabilities is beyond the reach of currently available methods. In fact, automatic evaluation of any generative model is a difficult task. Metrics commonly used for translation systems, such as BLEU [32], or for automatic summarization, such as ROUGE [33], are not appropriate for more sophisticated tasks such as explainability or even dialogue systems, since they assume that valid responses have significant word overlap with the ground truth responses [34].

For that reason, most evaluation methods for XAI systems require human intervention. For example, the organizers of a fake news detection competition ( https://leadersprize. truenorthwaterloo.com/en/ (accessed on 7 December 2019)), which requires an explanation of why a given statement is considered fake news or not, split the competition into two phases and limited the explanations assessment to the second phase to which only 10 teams would be qualified, thus making it manually tractable.

The history of evaluation in the field of data visualization is also relevant to the question of how to evaluate explanations. The initial focus on alternative visual renderings of data has, over a decade, transformed from whether a visualization was "interesting" to consideration for what human inferences are enabled by alternative visualization techniques (e.g., [35]).

The simplest conceptual alignment is that a visualization is a visual explanation. The compression of a large volume of data to a visualization picture is lossy and inductive, so the choice of how to create that lossy inductive picture or explanation is about what inferences are preserved for the human visual system. The evaluation of alternative visualizations has evolved to a framework where evaluation is about what inferences are easily observed (e.g., [36]). Furthermore, interactive visual explanation is easily considered as our suggestion of explanation being interactive and driven by the semantics of the application domain (e.g., [37]).

Evaluation of what we can consider as a more general explanatory framework, which produces alternative explanations in terms of text, rules, pictures, and various media, can similarly be aligned with the evolution of how to evaluate visual explanations.

Of course, there is yet no clear explanation evaluation framework but only a broad scope of important components (e.g., [9]). Even specific instances of proposals for explanation evaluation beg the need for increased precision. For example, Adiwardana et al. [30] suggest explanation quality is dependent on two main factors: sensibleness and specificity. They suggest a measure which takes those factors into account (Sensibleness and Specificity Average (SSA)). This suggestion arose from work on the topic of dialogue systems and has been characterized in terms of a high correlation with "perplexity", which is a measurement of how well a probability distribution or probability model predicts a sample. A low perplexity indicates the probability distribution is good at predicting the sample. In the context of conversational systems, perplexity measures the uncertainty of a language model, which is a probability distribution over entire sentences or texts. The lower the perplexity, the more confident the model is in generating the next token (character, subword, or word). Therefore, perplexity can be understood as a representation of the number of choices the model can select when producing the next token. This measure is commonly used to assess the quality of conversational agents and as a metric which must be optimized by machine

learning based dialogue models. Although not ideal and lacking specific experiments on the domain of explainability, perplexity could potentially be effectively used to evaluate text-based XAI systems as a reasonable approximation of human evaluation.

Finally, XAI systems are not limited to their value for humans but include their general role in assessment of predictive models. While a complete discussion is beyond our scope here, note the example of certifying the regulatory compliance of safety critical systems such as autonomous driving controllers. A regulatory agency will employ their certification software in an architecture that automates the interpretation of explanations that a learned autonomous controller predicts as safe actions.

While we have more to say about evaluation below, what is clear is that evaluation of explanatory systems is based on how the explainee confirms their own understanding of an explanation or the conclusion of an explanatory dialogue.

## 3. A Brief History of Explanation

### 3.1. Expert Systems and Abduction

Explanations have always been an indispensable component of decision making, learning, understanding, and communication in the human-in-the-loop environments. After the emergence and rapid growth of artificial intelligence as a science in the 1950s, an interest in interpreting underlying decisions of intelligent systems also proliferated. Initially, the AI community focused on developing expert systems [38,39], which were computer programs capable of providing human-level solutions to domain-specific problems in a variety of domains. The typical expert systems architecture used reasoning technique that employed if–then rule-based knowledge representation. These architectures focused on two main components: a domain-independent inference engine and a domain specific knowledge base. The knowledge base consists of application-specific data (e.g., facts and rules) to solve specific problems in the domain, and the inference engine applies these rules to infer new facts. However, by this design, expert systems were not effective to make decisions beyond provided domain knowledge, which was difficult and expensive to accumulate (i.e., the "knowledge acquisition bottle neck."), and were inferentially weak and unable to consistently capture complex operations and produce the reasonable rationale behind the decisions. In this architecture, C.S. Peirce's hypothesis of abduction [40] stimulated the AI community's attention to exploiting this conceptual framework for the design and development of complex intelligent systems in a variety of domains. By definition, abduction or abductive reasoning is a form of reasoning that starts with a set of observations and then uses them to find the most likely explanations for the observations. A compressed historical journey of Peirce's ideas can be traced in four projects, beginning with Pople [41], Poole et al. [42], Muggleton [43], to Evans et al. [26]. In 1973, Pople provided a description of an algorithm to implement abductive reasoning and showed its application to medical diagnosis. Poole et al. extended abductive ideas to a full first order implementation and showed its application to guide the creation of explanatory hypothesis for any application domain. Muggleton produced a further refined system called inductive logic programming, in which creation of hypotheses are generally identified by inductive constraints in any general logic. Finally, the adoption of this thread of mechanisms based on abductive reasoning has been generalized to the full scope of explanation generation based on inductive logic programming by Evans et al. Every instance of these contributions relies on a logical architecture in which explanations arise as rational connections between hypotheses and observations (cf. scientific explanation). The most recent work by Evans et al. extends the framework in a manner that supports modern heuristics of inductive model construction—or learning of predictive models—by providing the definition of a gradient measure to guide search over alternative inductive logic programs.

In another related alternative, Thagard [44] proposes a computational theory consisting of seven principles that create relations of explanatory coherence. According to the proposal, explanatory coherence can be thought of as (i) a relation between two propositions, (ii) a property of a single proposition, and (iii) a property of all sets of propositions

related to each other. Assuming S is an explanatory system and P, Q, and $P_1 \ldots P_n$ are its propositions, seven principles establish explanatory coherent system according to Thagard's proposal: (1) symmetry between propositions, (2) explanation capability of propositions themselves, (3) analogy between propositions, (4) data priority (i.e., proposition itself becomes acceptable on its own if it delineates the result of observation), (5) contradiction between propositions, (6) acceptability of proposition P in a system S, and (7) system coherence referring to pairwise local coherence of the system's propositions. Finally, he relates the concept to artificial intelligence by giving examples in probabilistic networks and explanation-based learning; for the latter one, he relates the concept of explanatory coherence to Peirce's hypothesis of abduction and concludes that his concept differs from abduction by being simpler and more explainable.

In fact, that thread of exploiting abduction in artificial intelligence is aligned with perspectives from other disciplines. For example, Eriksson and Lindström describe abductive reasoning as an initial step of inquiry to develop hypotheses where the corresponding outcomes are explained logically through deductive reasoning and experimentally through inductive reasoning [45]. Their application to "care science" is just another example that confirms the generality of abductive reasoning.

The block diagram of Figure 2, partially inspired by a figure in [46], is intended only to confirm the connection between abductive, deductive, and inductive reasoning. We see that abductive reasoning entails justification of ideas that support the articulation of new knowledge by integrating deductive and inductive reasoning. In artificial intelligence studies, the process involving these reasoning steps are as follows: (1) identify observations that require explanation as they cannot be confirmed with already accepted hypotheses; (2) identify a new covering hypothesis using abductive reasoning; (3) empirical consequences of the hypothesis, including consistency with already known knowledge, is established through deduction; and (4) after an accepted level of verification, the hypothesis is accepted as the most scientifically plausible.

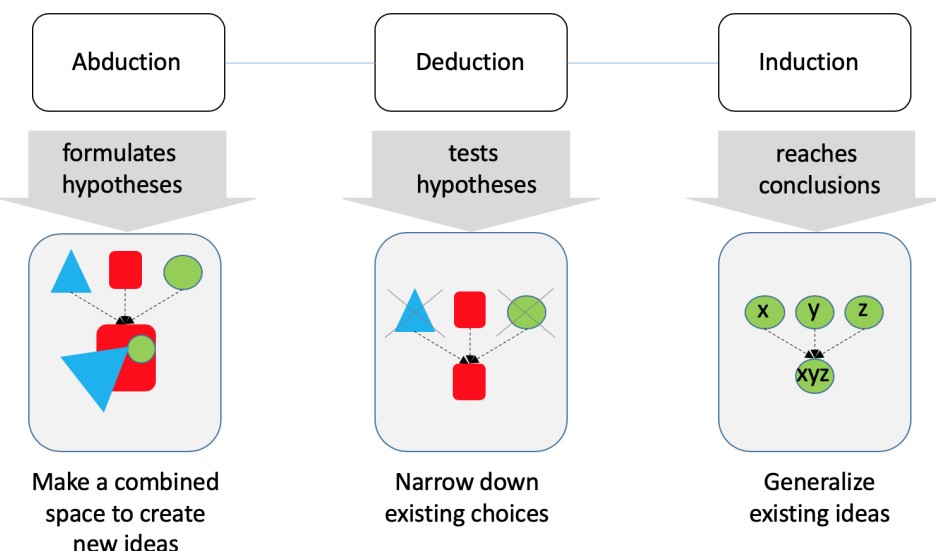

**Figure 2.** The process steps of the reasoning methods.

It is also noteworthy to mention that humans do not always need explanations of intelligent systems. For instance, it is generally understandable why a GPS navigator recommends a specific route while driving and humans will probably not require explanations in this context. Another example, when users search for specific topics on search engines, they may see suggestions for similar content and it is clear that that content is suggested based on the similarity and pertinence to the initially searched keywords. Explanations become essential when actions or decisions made by such systems are critical, unclear, and not easily understandable. These explanations need to follow the principles of scientific or,

at least, causal explanations (if a user is satisfied with a simple explanation of an event), overviews of which are provided in the next subsections.

### 3.2. Scientific Explanation

The connection between artificial intelligence frameworks for abductive explanation have suggested a direct connection between "scientific explanation" and is the subject of many debatable issues in the community of science and philosophy [5]. Some of the discussions imply that there is only one form of explanation that may be considered scientific. Some proponents suggest that a theory of explanation should include both scientific and other simpler forms of explanation. Consequently, it has been a common goal to formulate principles that can confirm an explanation is a scientific explanation. This idea is as old as Aristotle, who is considered to be the first philosopher to articulate an opinion that knowledge becomes scientific when it tries to explain the causes of "why". His view urges that science should not only keep facts but also describe them in an appropriate explanatory framework [47].

In addition to this theoretical view, empiricists also maintain a belief that the components of ideas should be acquired from perceptions with which humans become familiar through sensory experience. The development of the principles of scientific explanation from this perspective prospered with the so-called deductive-nomological (DN) model that was described by Hempel in [48–50] and by Hempel and Oppenheim in [51].

The DN model is based on the idea that two main elements form a scientific explanation: an explanandum, a sentence that outlines a phenomenon to be explained, and an explanan, a sentence that is specified as explanations of that phenomenon. For instance, one might constitute an explanandum by asking "Why did the dish washer stop working?" and another person may provide an explanan by answering "Because the electricity went off." We may infer that the explanan is rationally or even causally connected to the explanandum, or at least that the explanandum is the reasonable consequence of the explanans, otherwise speaking [52]. In this way, the explanation delivered as an explanans becomes a form of deductive argument and constitutes the "deductive" part of the model. Note that a series of statements comprising an explanan should comply with "laws of nature." This is a vital property because derivation of the explanandum from the explanan loses its validity if this property is violated [5]. This is the nomological component of the model, where the term "nomological" means "lawful." Hempel and Oppenheim's DN model formulation states that a scientific explanation is an answer to a so-called "why" question and there may be multiple such answers. There may also be several types of questions (e.g., "How does an airplane fly?") that cannot be converted into "why" questions. Consequently, answers to such questions are not considered to be scientific explanations. This does not mean that such answers are not part of a scientific discipline; these answers just become descriptive rather than being explanatory [53], and this is related to our next section on causality.

Another aspect of the DN model is that the elements of an explanation are statements or sentences describing phenomenon, not the phenomenon itself. Finally, the sentences in the explanans must be accurate and verified, urging the arguments of the scientific explanation to be valid (i.e., providing conclusive support) and sound (i.e., having true premises). Thus, the DN model can be summarized as a model of a scientific explanation outlining a conception of explanation and a connection in the flow of an explanation.

### 3.3. Causality

As hinted in the summary of scientific explanation, perhaps the most strict form of explanation is causal explanation. Informally, a causal explanation is one that arises from the construction of causal models, which require that explanations for arising predictions are, in fact, "recipes" for reconstructing that prediction.

Causal models typically facilitate the creation of explanation for a phenomena or an answer to a query by constructing a formal expression. That formal expression is derived from some causal representation, which typically captures directed causal relationships,

perhaps most recently in a graphical model of cause and effect (or causality) (cf. [6]). This representation encodes an incomplete set of assumptions built upon prior knowledge. The causal explanation expression is, as with abduction and scientific explanation, revised continuously until a suitable explanation is obtained.

The most relevant and recent framework of causal representation and reasoning is given by the culmination of Pearl's research in [6]. In that work, an abductive explanation is called an "estimand." This idea of a formal expression that best explains a particular phenomena (or a query) has its root in formal philosophy, as noted about, and especially in, abductive reasoning. As noted above, abductive reasoning is given a set of incomplete observations (or assumptions, as described above) and seeks to construct an explanation which best describes it (or the estimand).

An important point here is that the overall information architectures of abduction, scientific reasoning, and causal reasoning are similar, but their mechanism and the evaluation of an explanation are successively refined.

### 3.4. Explaining Mechanism/Syntax Versus Semantics

A lingering unaddressed distinction is about the content or meaning of an explanation, especially in the context of what counts as an explanation for an explainee, the target of an explanation. Again, a principled distinction exists in the realm of mathematical logic (cf. [19]; any logic textbook will suffice). In the context of predictions from domain models (whether learned or fabricated by hand), a prediction has at least two kinds of explanation. For example, consider the simple familiar syllogism:

- All men are mortal.
- Socrates is a man.
- Socrates is mortal.

Consider "Socrates is mortal" as a prediction of the very simple model. From the perspective of formal logic, there are (at least) two explanations. One is the explanation of the deductive mechanism that produced "Socrates is mortal" from the first two expressions. This is a so-called proof-theoretic explanation as it amounts to a description of how two premises are combined by deductive inference to derive the prediction. In an analogy with programming language debuggers, this kind of explanation is about the mechanism that produced the prediction and is akin to current work in explaining image classification (e.g., [21]). This kind of explanation is appropriate when the explainee has interest in understanding and debugging the mechanism.

However, note an alternative explanation is not about mechanism but about the meaning of the expressions. Logically, the proof theory or deductive chain explanation is about mechanism. However, the semantic explanation is about what it means to be mortal and what it means to be a man. That kind of explanation is semantic and is intended to be appropriate for an explainee who is not interested in mechanism but in meaning. If a prediction was "Socrates is a duck," obtained from the same system, it can immediately be viewed with suspicion because of its meaning, not because of the mechanism that produced it from a presumably faulty model.

So, distinguishing syntax from semantics or meaning has more to do with the internal rules that a system has to follow to compute something. We all know that symbolic debuggers for programming languages create labels and traces which become the basis for producing mechanism explanations. The computation rules themselves might not be sufficient to provide a clear picture on why a system came to a conclusion (or produced an answer to a query). However, the systematic combination of those rules creates the foundation of semantic interpretation of the query or explanation. Returning to Pearl's nomenclature of estimands, an estimand can be viewed as a well-constructed expression if it makes sense semantically. As with the simple syllogism above, the form of the explanation can be based on that of the causal (or any) model.

In this era of deep learned models, we can consider these relationships between syntax and semantics as the internal representations of each layer and their composition at the

final layer, respectively. Interestingly, this notion of construction of semantics (whole) as a function of semantics of its parts and their careful combination that obeys a particular syntax is very familiar in the logics used to interpret natural language, developed by a famous linguist logician, Richard Montague [54]. At the syntactic level, we might infer the correlation among different variables (in the intermediate layers) of a deep learned system, but in the semantic level we know what combination of those variables (in the final layer) provide an interpretation for a particular query. An ontology-driven explanation for a ResNet model [24], described in Section 2, is one good example of the use of semantics to explain an opaque system.

## 4. Classification of Research Trends Based on Levels of Explanation

In the last five years, there has been a surge in papers that attempt to introduce new explanation methods. This intensity of work in XAI is, in fact, a side effect of widespread use of AI in sensitive domains such as legal reasoning and the medical field. In this section, we consider a few of the explanation approaches in the literature and classify them based on how/when the explanations are built (post hoc vs. concurrently constructed), and compare that with the kinds of explanation introduced in Section 1. Note that this is not an exhaustive list of methods, yet they act as examples for describing the classification. In addition, XAI techniques are used in many applications such as anomaly detection [11], traffic analysis [14], and neural networks [12,13].

### 4.1. Concurrently Constructed Explanations

Some authors have focused on creating models that try to build explanations concurrently together with the main task (e.g., learning a classifier). As an example, consider the work of [55], who seek to identify segments of text that support an explanation of text reviews classification. Their approach proposes a neural architecture that is made up of a generator (which is a "rationale" extractor), followed by an encoder component (which is a classifier). The generator extracts portions of the input text as salient words, then forwards them to the encoder to predict the target class. The final output of the system comprises the class label, together with the extracted "justification" from the input text.

As another example, in natural language processing (NLP) text classifications, an elaboration uses attention mechanisms to attempt to learn the weight of each word within an identified rationale or justification (a subset of the text being classified) [56]. There is a debate in the literature on whether attention weights could be used as an explanation or not [57,58]. However, this discussion is beyond the scope of this work. However, there is an interesting connection to our discussion above regarding the difference between debugging explanations and semantic explanations; much of this research is motivated to identify mechanism behaviour to semantic interpretability.

### 4.2. Post-Hoc Explanations

Another approach is to use a post-hoc technique, which attempts to build a representation of salient predictive model components after that model is produced. Simply put, the idea is to approximate the content of explanations from a trained model. As mentioned earlier, concurrently constructed explanations need to be computed within the model, which means they need to have access to the internals of the model, or what many refer to as "model dependent" (this further creates confusion about whether a model is syntactic or semantic). However, some post-hoc approaches can create approximate explanations without having access to the internals of the model, and thus could be classified either as model dependent or model independent (sometimes these are called model-agnostic post-hoc explanations). In the next subsection, we will briefly discuss the difference between model-dependent and model-independent explanation mechanisms.

### 4.2.1. Model-Dependent Explanations

To describe model-dependent explanations, consider the case of non-linear deep networks. One can use a back-propagation algorithm to learn feature importance (e.g., which pixels contributed most in classifying the image as a cat rather than a dog), then use that learned feature ranking as the basis for explaining predictions. The simplest general approach is to compute a gradient with respect to the predicted class and use the back propagation to propagate the gradient to the input. Finally, one can combine the input with the gradient to capture the salient pixels which can be used to explain the predicted class (e.g., Grad-CAM [59]).

### 4.2.2. Model-Independent Explanations

The goal of this group of methods is to focus more on explaining individual instances without the target model being exposed. In fact, the target model is now a black-box model. Ribeiro et al. introduced LIME [60] to approach the explanation problem using a perturbation method. They perturb the original data point to create a new dataset in the vicinity of that instance. The black-box model is queried to obtain the labels associated with the aforementioned points. This labeled dataset is then used to frame a near enough justification. While LIME is the most cited model-independent method, there are other approaches which can be classified as model independent [61,62].

### 4.3. Application Dependent vs. Generic Explanations

Another way to classify explanation methods is to consider how an explanation mechanism is related to the application domain of the task. An application-dependent method implicitly assumes the explainee is knowledgeable about the application area, and thus it employs the domain's vocabulary. In a medical application, for instance, a system can explain the prediction using medical terms. A generic explanation, on the other hand, can only provide explanations based on the mechanism of model building, combined with information available in the training set (e.g., correlation between features). Note that a model-dependent method is not necessarily considering the knowledge of the explainee (i.e., it will provide the same explanation regardless of the explainee's knowledge), but it must take advantage of the application's vocabulary (see Section 2.3). It is also noteworthy that this XAI system design needs to go beyond correlative features—which is how most current machine learning methods work—to be capable of providing such application-dependent explanations. Many explainees (e.g., physicians, lawyers) would prefer having application-dependent explanations. This will not be achieved without moving the machine learning research on explanation toward scientific and causal explanation.

### 4.4. Classification Based on Levels of Explanation

As described briefly in Section 1, different levels of explanation could be classified in a number of ways, including by whether they are model dependent (MD), model independent (MI), and created concurrently (CC) or in a post-hoc (PH) fashion. Here, we want to further elaborate those abstract levels and classify the related work accordingly. Table 2 classifies some of the most prominent existing work based in this way.

**Table 2.** Classification of recent explanation techniques based on levels of explanation. MD stands for model dependent while MI means model independent. CC corresponds to concurrently constructed explanations and PH refers to post-hoc technique.

| Method | Level | MD/MI | CC/PH |
|---|---|---|---|
| LIME [60] | 1 | MI | PH |
| Grad-CAM [59] | 1 | MD | PH |
| SHAP [61] | 1 | MI | PH |
| Rationalizing predictions [55] | 1 | MD | CC |
| Grounding visual explanations [63] | 2 | MD | PH |

Another, perhaps more informative assessment of XAI system capabilities can be conducted using not only the properties noted in Table 1 but the degree to which any XAI system provides a combination of those capabilities, as described in Figure 3. Even without more precision on these major aspects of XAI systems, the figure serves to separate five classes of XAI architectures (including level 0 and four others). We acknowledge that no consensus yet exists to precisely define the separation of these levels, for example, recording the name of an explainee is not necessarily of value in customizing an explanation, but their history of knowledge about mathematical logic would be. The point is only that including some instance of each component provides a basis for a "higher level" XAI system.

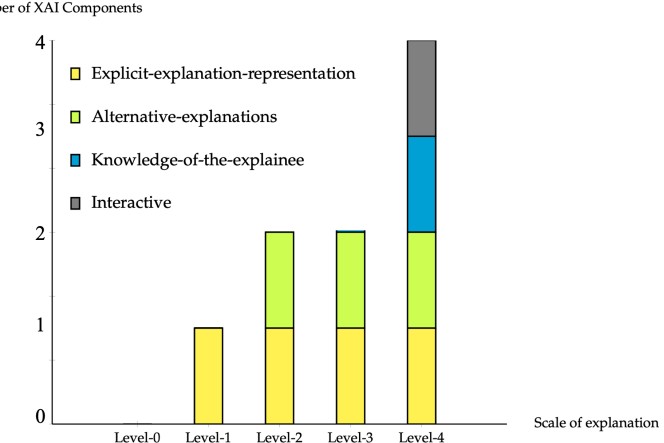

**Figure 3.** Major explanatory components (stacked bar) and their potential role in a scale of explanation.

### 4.4.1. Level 0: XAI Application Provides No Explanation

Models classified as Level 0 provide no explanation at all. They are, in essence, black-box models that do not provide any explanatory information to a user. In other words, the explainee is expected to accept or reject a system's prediction without any further information. Most off-the-shelf methods for learning classifiers (e.g., deep learned models, support vector machines, or random forests) belong to this level. Note also that any predictive system, whether based on machine-learned predictive models or not, is in this category.

### 4.4.2. Level 1: XAI Application Provides Explicit Explanations

The explainee is provided with a single type of explanation in models falling into this category. For example, a framework that provides heat maps to explain image classification belongs to this level. Most of this approach focuses on providing a post-hoc explanation, which moves a black-box model—that originally belonged to Level 0—to a Level 1 model. Recently, however, a few methods have been proposed to look at building concurrently constructed explanation algorithms [55,64] to make models that, by our definition, belong to Level 1.

### 4.4.3. Level 2: XAI Application Provides Alternative Explanations

Level 2 adds another enrichment of explanation to improve knowledge communicated with the explainees. At this level, for example, an image classification explanation system could provide not only a heat map to explain a classified animal image as a cat, but it could also provide another type of explanation such as a textual explanation that notes a predictive model's bias to a category (e.g., bias to cats) as an alternative description of the predicted classification. In this way, the alternative explanations allow the explainee to grasp further insight about the reasoning process employed by the system to make the prediction. If one explanation is not well understood by the explainee, they have the

opportunity to understand from an alternative explanation. Note that in the case of the abductive systems described above, there can be a large number of alternative explanations.

### 4.4.4. Level 3: AI Application Provides Explainee-Aware Alternative Explanations

An explainee and their familiarity with an application domain plays a vital role in this level. This level of explanation of XAI system, in addition to providing multiple explanations, includes some model of the explainee's domain knowledge and is capable of deciding an appropriate type of explanation according to the knowledge of the explainee. For instance, consider when a patient is diagnosed with some disease and an AI system is used to provide a potential treatment therapy. While the therapist requires a detailed medical explanation by the AI system, the patient would strongly prefer to have a lay person's explanation for any alternative treatment recommendations.

In the current context of the COVID-19 pandemic, as an example, Hydroxychloroquine is alleged to be a potential cure and has attracted many ordinary people's attention around the globe. People are interested to understand why this drug is a potential treatment. As a result, many medical researchers provide interviews to the media explaining how this drug works, typically with very shallow detail. As we can expect, however, the same experts would use a different level of granularity to explain the drug to other experts. Please note that none of the existing explanation methods take into account the knowledge of the explainee.

### 4.4.5. Level 4: XAI Application Interactively Provides Explainee-Aware Alternative Explanations

While previous levels (e.g., Level 0, 1, 2, 3) do not include the capability of interacting with an explainee, except perhaps for requesting alternative targeted explanations (Level 3), XAI methods classified as Level 4 can interact with the user. They are expected to support a conversation capability which allows the explainee to refine their questions and concerns regarding a prediction. In other words, each interaction in the conversation allows the explainee to obtain clarifications. Here, the system is capable of adapting its explanation to the dynamic information provided by interaction with the explainee. Take the Richard Feynman interview [23] with the BBC as an example. He could provide the reporter with what he thought the reporter would understand most. Once the reporter understood that explanation, if he had further questions, or wanted more in-depth explanation, the reporter could ask, and an appropriate explanation could be provided by Richard Feynman. To the best of our knowledge, existing systems lack this interaction capability.

## 5. Priority Components for a Synthesis of an XAI Architecture

### 5.1. XAI Architecture

As noted, much of the work on the explainability has focused on deep supervised learning, which describes methods that answer the following two questions: (1) which input features are used to create an output prediction and (2) which input features are semantically correlated with the outcome prediction. The answers for these two questions contribute to the trust in the system, but explanation additionally requires a social process of transferring knowledge to the explainee considering their background knowledge.

While the answers to questions (1) and (2) may acknowledge the importance of features that a model uses to arrive at a prediction, it may not necessarily align with a human explanation; prior knowledge, experience, and other forms of approximate reasoning (e.g., metaphorical inference) may further shape an explanation, while the predictions of a machine learning model may be restricted to the dataset and the semantics around it. Generally, an explanation system (for example, a human) is not restricted to the knowledge on which they make predictions and explanations and can draw parallels with different events, semantics, and knowledge.

So, merely responding to questions (1) and (2) will not satisfy the multiple purposes that XAI researchers aim to achieve: to increase societal acceptance of algorithmic deci-

sion outcomes, to generate human-level transparency about why a decision outcome is achieved [65], and to have a fruitful conversation among different stakeholders concerning the justification of using these algorithms for decision making [66].

To incorporate an interactive "explainer" in XAI, an emerging XAI architecture needs to embed both an explainable model and an explanation interface. The explainable model includes all types of the pre-hoc, post-hoc, and concurrent explanation models. As examples of the explainable model, there can be a causal model, an explainable deep adaptive program, an explainable reinforcement learning model, etc. An explanation interface can also be a variety of types, such as a visualization system or a dialogue manager with a query manager and a natural language generator that corresponds to Level 3 and Level 4 of Figure 3.

### 5.2. User-Guided Explanation

As Miller [9] notes, the process of explanation involves two processes: (a) A cognitive process, namely the process of determining an explanation for a given event, called, as with Hempel, the explanandum. This identifies causes for the event, and a subset of these causes are selected as the explanation (or explanans). (b) A social process of transferring knowledge between explainer and explainee, generally an interaction between a group of people, in which the goal is that the explainee has enough information to understand the causes of the event. This is one kind of blueprint for the Level 4 interactive explanation process noted above.

Miller also provided an in-depth survey on explanation research in philosophy, psychology, and cognitive science. He noted that the latter could be a valuable resource for the progress of the field of XAI and highlighted three major findings: (i) Explanations are contrastive—people generally do not ask why event *E* happened, but rather why event *E* happened instead of some other event *F*; (ii) Explanations are selective and focus on one or two possible causes and not all causes for the recommendation; and (iii) Explanations are social conversation and interaction for transfer of knowledge, implying that the explainer must be able to leverage the mental model of the explainee while engaging in the explanation process. He asserted that it is imperative to take into account these three points if the goal is to build a useful XAI.

One should note that it is plausible, given the study of explanation based on cognitive norms, that an explanation may not be required to be factual but rather only to be judged to be satisfactory to the explainee (cf. Sections 2.5 and 4.3).

As we described in Figure 3, a dialogue system that can process a question of "What about another condition" from an explainee and produce a new prediction output based on the new condition will achieve another higher level of explanation. The explanation that can deal with "What would the outcome be if the data looked like this instead?" or "How could I alter the data to achieve outcome X?" is called contrastive explanation. Contrastive explanation is considered to be a human-friendly explanation as it mimics human explanations that are contrastive, selective, and social.

To accommodate the communication aspects of explanations, several dialogue models have been proposed. Bex and Walton [10] introduce a dialogue system for argumentation and explanation that consists of a communication language that defines the speech acts and protocols that allow transitions in the dialogue. This allows the explainee to challenge and interrogate the given explanations to gain further understanding. Madumal et al. [67] also proposed a grounded, data-driven approach for explanation interaction protocol between explainer and explainee.

### 5.3. Measuring the Value of Explanations

The production of explanations about decisions made by AI systems is not the end of the AI explainability debate. The practical value of these explanations, partly, depends on the audience who consumes them: an explanation must result in an appropriate level of understanding for the receivers of explanations. In other words, explanations are required

to be interpreted and judged by multiple criteria, about whether they are good or bad, satisfactory or unsatisfactory, effective or ineffective, acceptable or unacceptable, and whether they are sufficient to inform decision making.

Again, the previously mentioned evolution of the evaluation of visualization systems is highly relevant as that evolution ultimately requires the design of cognitive experiments to confirm the quality and value of alternative explanations, visual or not (see Section 2.5). It is clearly the case that quality of a "visual explanation" is about how well it leads the reader to the intended inferences from the visualized data domain. Naturally, the background knowledge of a viewer is like the background knowledge of an explainee; their knowledge and experience determines what preferred inferences obtain.

Looking forward to how to evaluate XAI systems, among those background assumptions that impact the judgements of explanations are what are returned to as cognitive "norms." It has been empirically shown that norms influence causal judgements [68]. To put it simply, norms are informal rules that are held by people and can have statistical or prescriptive content. The empirical and mathematical aspects for why a decision outcome is achieved are interpreted against some background assumptions held by the assumed audiences of explanations. Some disagreements with an explanation for a decision outcome in a sensitive context due to the background assumptions of the audience of explanations reveal some moral or social mismatch about algorithmic decision making between the receiver of an explanation and its producer.

If one does not have an appropriate level of knowledge about the relevant precedent assumptions, one might not have the capacity to judge and interpret an explanation of a decision. In that case, iteratively refined question–answer dialogue (cf. Feynman's point made in Section 2.2) may lead to an improved understanding by the explainee. In general, the interpretability of explanations has significant practical value for revealing the explicit and the implicit reasons about why a decision-making procedure and process is chosen.

A schema for the interpretability of explanations aims to capture various precedent assumptions that become relevant in context-dependent evaluation of each kind of AI explanation for why a decision outcome is achieved.

Finally, another elaboration of how to evaluate explanations [69] proposes five measures of explanation effectiveness: (1) user satisfaction, (2) mental model, (3) task performance, (4) trust assessment, and (5) correctability.

User satisfaction is measured in terms of clarity of the explanation and utility of the explanation. Task performance is the idea of checking that an explanation improved a user's decision and task performance. Trust assessment is to assess trust and suggest that an XAI system is appropriate for future use. Assessment of a mental model is related to strength/weakness assessment, and it also assesses the predictions of "what will it do", or what-if questions, and "how do I intervene" to adjust or guide explanatory outputs. Finally, correctability is intended to measure if interaction with the system helps to identify and correct errors. As far as we are aware, there is also no Level 4 system that has confirmed any experiments that demonstrate this kind of richly faceted evaluation.

Finally, to measure contrastive explanation that is close to human explanation, we need additional evaluation metrics for contrast, selection, and social explanation. Contrast can be measured in terms of the clear justification of the output through the comparison. Contrastive explanation should be able to explain why the output has been produced between the probable output candidates. Selection can be measured in terms of the importance (salience) of the reasons (features) that were mentioned during the contrastive explanation. Lastly, social explanation can be measured in terms of the clarity, understandability, and utility of the explanation to the explainee. The measure of the social explanation corresponds to the measure of user satisfaction in [69]. However, as noted, we know of no existing explanation systems that have been so considered with this rich palette of evaluation parameters.

## 6. Summary and Conclusions

Our goal has been to identify a plausible set of required components of an XAI architecture and describe a high-level framework to understand their connections. In two alternative graphical depictions (Figure 3, Table 1), we distinguish what we believe are mostly orthogonal components of an explanation system and suggest an information framework related to levels of autonomous driving, where a richer set of components provides a more sophisticated explanation system.

That framework is descriptive and informal, but it allows us to factor some components (e.g., interpretability, explanation quality) into separate analyses, which we hope creates some line of sight to historical work on explanation. Nowhere is this more important than the history of abductive reasoning and its connection to the history of scientific reasoning, which culminates in the construction and use of causal models as a basis for causal explanations.

We then try and consider more recent research in the context of these components and their relationship to the analysis of a deeper background literature and provide some description of how those early ideas fit, and what they lack. This concludes with a sketch of how to evaluate explanatory systems and connects recent work that addresses the cognitive properties of explanations. Overall, we hope that our framework and analysis provide some connective tissue between historical threads of explanation mechanisms and modern reinterpretation of those mechanisms in the context of cognitive evaluation.

We conclude that there is much still to achieve to inform a principled design of a high-level explanation system, but that there are many components and integrating them with the appropriate knowledge representations within machine learning models, and respecting the cognitive aspects of evaluation, are a minimal requirement for progress.

**Author Contributions:** Conceptualization, R.G.; funding acquisition, R.G.; investigation, M.-Y.K., S.A., H.K.B.B., N.F., R.G., O.R.Z., M.-H.M., J.R., T.S., H.Y. and P.C.; methodology, R.G.; supervision, M.-Y.K. and R.G.; visualization, S.A., H.K.B.B., O.R.Z. and J.R.; writing—original draft, M.-Y.K., S.A., H.K.B.B., N.F., R.G., O.R.Z., M.-H.M., J.R. and T.S.; writing—review and editing, M.-Y.K., R.G., O.R.Z. and H.Y. All authors have read and agreed to the published version of the manuscript.

**Funding:** This research was funded by University of Alberta's Alberta Machine Intelligence Institute (Amii).

**Acknowledgments:** This research was supported by University of Alberta's Alberta Machine Intelligence Institute (Amii).

**Data Availability Statement:** All data are contained within the article.

**Conflicts of Interest:** The authors declare no conflict of interest.

## Abbreviations

The following abbreviations are used in this manuscript:

| | |
|---|---|
| AI | Artificial Intelligence |
| XAI | Explainable Artificial Intelligence |
| CNN | Convolutional Neural Network |
| BBC | British Broadcasting Corporation |
| ILP | Inductive Logic Programming |
| SSA | Sensibleness and Specificity Average |
| GPS | Global Positioning System |
| DN | Deductive Nomological |
| NLP | Natural Language Processing |
| LIME | Local Interpretable Model-Agnostic Explanations |
| MD | Model Dependent |
| MI | Model Independent |
| CC | Concurrently Constructed |
| PH | Post Hoc |

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
