# Peer review of "A Multi-Component Framework for the Analysis and Design of Explainable Artificial Intelligence"

_make, doi:10.3390/make3040045_

Round 1

Reviewer 1 Report

Question 01:  The contribution and the originality of the proposed work need to be clarified at the beginning of the paper. This paper can be rather considered as a review.

Question 02: The introduction should be improved by adding some recent references dealing with the subject. The authors should present a more detailed description of the existing approaches

Question 03:  How could we connect your analysis with the level of confidence (or score confidence) that we can attribute to a model. How can we reach a certified AI

Question 04 : The section  “4.4. Classification based on levels of explanation” describing level AI needs absolutely to be illustrated by more references. Additional examples are also necessary in this section.  

Reviewer 2 Report

The authors provide a theoretical framework of current XAI components, describing XAI requirements and giving details on their connection to the history of XAI ideas. Moreover, the authors provide a survey of recent XAI-related literature and methodologies.

In my opinion, the paper still needs moderate revision before possible publication in the MDPI Machine Learning and Knowledge Extraction journal.

More detailed comments are provided in the following.

1) The paper should be carefully proofread to remove grammar errors and typos.

2) Figure 5 should be completely re-drawn. Indeed, the legend is unreadable (the font is too small) and the meaning of the stacked bar chart is not clear. Which is the unity of measurement of the y-axis?

3) Given its partially technical nature, albeit interesting, the paper is hard to follow. Therefore, the authors should provide guidelines dependent on the prior knowledge and expertise of the reader regarding XAI methodologies.

4) The authors should consider practical applications of XAI techniques. A non-comprehensive list of related works that should be taken into account is the following:
[R1] https://doi.org/10.1145/3359992.3366639
[R2] https://doi.org/10.1109/TNSM.2021.3098157
[R3] https://doi.org/10.1145/3387514.3405859
[R4] https://doi.org/10.1109/HSI.2018.8430788

Reviewer 3 Report

This paper presents a framework for the analysis and design of explainable artificial intelligence (XAI) by providing a strategic inventory of its requirements and highlighting the connections between them and the history of XAI requirements. As a final contribution, this work allows synthesizing the ideas of XAI requirements into a framework to inform the design and evaluation of the XAI system.

The paper is interesting, but in my opinion, it lacks an adequate structure. Also, the document is hard to read, and it needs to be reviewed to highlight its main contribution. In addition, it is necessary to compare the proposal with other similar works. Finally, it would be appreciated to represent certain information visually(e.g, figures, tables).

Round 2

Reviewer 3 Report

Thanks to the authors for your gentle response. I agree with them that this topic is hard to be addressed. In this sense,  the introduction meets its goal of preparing the reader to understand the rest of the connections between history and current vocabulary.

The paper was improved from its previous version. From my viewpoint, the document doesn't require more changes.